# Offline Bayesian Aleatoric and Epistemic Uncertainty Quantification and Posterior Value Optimisation in Finite-State MDPs

**Filippo Valdettaro**[1]

**A. Aldo Faisal**[1,2,3]

[1]Brain & Behaviour Lab, Dept. of Computing, Imperial College London, UK
[2]Brain & Behaviour Lab, Dept. of Bioengineering, Imperial College London, UK
[3]Chair in Digital Health & Data Science, University of Bayreuth, Germany

## Abstract

We address the challenge of quantifying Bayesian uncertainty and incorporating it in offline use cases of finite-state Markov Decision Processes (MDPs) with unknown dynamics. Our approach provides a principled method to disentangle epistemic and aleatoric uncertainty, and a novel technique to find policies that optimise Bayesian posterior expected value without relying on strong assumptions about the MDP's posterior distribution. First, we utilise standard Bayesian reinforcement learning methods to capture the posterior uncertainty in MDP parameters based on available data. We then analytically compute the first two moments of the return distribution across posterior samples and apply the law of total variance to disentangle aleatoric and epistemic uncertainties. To find policies that maximise posterior expected value, we leverage the closed-form expression for value as a function of policy. This allows us to propose a stochastic gradient-based approach for solving the problem. We illustrate the uncertainty quantification and Bayesian posterior value optimisation performance of our agent in simple, interpretable gridworlds and validate it through ground-truth evaluations on synthetic MDPs. Finally, we highlight the real-world impact and computational scalability of our method by applying it to the AI Clinician problem, which recommends treatment for patients in intensive care units and has emerged as a key use case of finite-state MDPs with offline data. We discuss the challenges that arise with Bayesian modelling of larger scale MDPs while demonstrating the potential to apply our methods rooted in Bayesian decision theory into the real world. We make our code available at `https://github.com/filippovaldettaro/finite-state-mdps`.

## 1 INTRODUCTION

In safety-critical machine learning applications, accurately quantifying confidence and uncertainty in decision outcomes becomes imperative for regulatory and trust reasons [Chua et al., 2022, Kendall and Gal, 2017]. Uncertainties that such systems face can stem from limited data availability (epistemic) or originate from inherent environmental randomness (aleatoric). Uncertainty quantification is particularly relevant and challenging in reinforcement learning (RL) systems as uncertainty in decisions' outcomes compounds in sequential decision-making.

We utilise inference schemes from classic Bayesian RL to account for epistemic uncertainty, with an exact-inference Bayesian dynamics model that assigns posterior probabilities to environments Duff [2002]. Aleatoric uncertainty is addressed by exploiting the analytic solutions to the linear equations for higher return distribution moments [Sobel, 1982]. Our contribution on the uncertainty quantification side is combining these two ingredients to determine overall aleatoric and epistemic standard deviations. We consider the computation and accuracy tradeoff of our method with prior work that does not exploit the tractability of finite-state MDPs.

On the control under uncertainty side, we propose a novel stochastic gradient-based method for policy optimisation that accounts for model dynamics uncertainty by optimising a policy for value under the environment posterior. In contrast to previous methods [Delage and Mannor, 2010], we do not rely on strong assumptions about the posterior distribution. We empirically demonstrate its performance and scalability, providing results on gridworlds and synthetic MDPs with varying offline dataset sizes, where we observe benefits in MDPs with higher uncertainty and lower data. Finally, our methods finds application in clinical decision support systems (CDSS), which leverage vast patient datasets to train RL algorithms for treatment suggestions [Gottesman et al., 2019, Li et al., 2020]. We analyse a setup used for sepsis treatment [Komorowski et al., 2018], where

patients' condition and treatment options were clustered into finite states and actions, originally tackled by applying dynamic programming methods that assume known environments [Bellman, 1957]. The methods presented enhance this approach with uncertainty quantification and uncertainty-aware control. We investigate the scalability of our methods in such practical environments and investigate the difference in expected posterior value between ours and the original policy optimisation method proposed. For this analysis, we included pessimism in the face of uncertainty, a common and necessary ingredient in offline RL [Kidambi et al., 2020, Yu et al., 2020, An et al., 2021] especially when the dataset does not adequately span the full state-action space [Agarwal et al., 2020], in the form of a conservative dynamics model.

## 2 RELATED WORK

This section reviews uncertainty treatment in finite-state MDPs. We focus on epistemic uncertainty in Robust and Adaptive MDP settings, aleatoric uncertainty for risk-averse policies, and recent work quantifying both types of uncertainty in finite-state offline decision making contexts.

**Robust and Adaptive MDPs.** A simple model-based approach for an MDP uses relative visitation frequencies as the ground truth transition probabilities. This can introduce bias and result in policies that generalise poorly [Mannor et al., 2007, Wiesemann et al., 2013, Chow et al., 2015]. To address this, a Bayesian approach is often employed to account for uncertainty in ambiguous transition dynamics, a common method in Bayesian RL [Ghavamzadeh et al., 2015]. Bayesian dynamics models used in Bayes-Adaptive MDPs (BAMDPs) [Duff, 2002, Guez et al., 2012, Rigter et al., 2021] maintain the current belief in transition dynamics and enable optimal 'offline' planning of adaptable 'online' policy rollouts. However, these models may be intractable beyond simple MDPs [Poupart et al., 2006, Lee et al., 2018, Zintgraf et al., 2021].

In high-risk offline settings, exploration is undesirable. For instance, in the CDSS developed in [Komorowski et al., 2018], novel actions are avoided by only considering actions above a minimum visitation threshold. Therefore, we focus on optimal *memoryless*, stationary (non-adaptive) policies depending only on the state [Delage and Mannor, 2010]. Finding policies that are robust to the worst-case realisation of uncertain dynamics can often lead to overly conservative policies, making average value optimization across a distribution of MDPs a better alternative [Nilim and Ghaoui, 2003, Iyengar, 2005, Xu and Mannor, 2006], and a principled one in line with Bayesian decision theory Robert et al. [2007]. This will be the problem formulation we will be tackling here, while requiring that our methods scale to medium-sized (approximately $10^3$ states) MDPs in data regimes where significant uncertainty is still present, in order for them to be applicable to real-world tasks. Delage and Mannor [2010] proposes a gradient-based method to optimise this objective, but makes the strong assumption that higher moments of the posterior distribution of transition parameters are small. Dimitrakakis [2011] proposes an algorithm that provides provably near-Bayes-optimal stationary policies, but focuses on providing an expected utility bound with respect to the Bayes-optimal *adaptive* policy, which in general has different utility to the optimal offline stationary policy.

**Risk-averse policies.** Accounting for inherent environmental stochasticity is often desirable. Using the distributional RL framework [Bellemare et al., 2017], policies are often informed by return distribution properties other than its mean to select risk-averse actions [Dabney et al., 2018, Clements et al., 2019]. However, optimal policies for such statistical functionals are generally neither memoryless nor time-consistent [Sobel, 1982, Bellemare et al., 2023]. Therefore, we focus on using the mean of the return distribution to guide the agent's policy.

**Aleatoric and Epistemic Uncertainty in Finite-state MDPs.** Several recent efforts have tried to model both types of uncertainties in discrete environments. In a healthcare context, Joshi et al. [2021] used a Bayesian dynamics model and Monte Carlo trajectory sampling to estimate uncertainties and determine when to defer treatment. In contrast, Festor et al. [2021] trained an ensemble of distributional deep neural networks (DNNs) to learn the return distribution, effectively learning a 'distribution over return distributions'. Neither of these works exploit the benefits of having a stationary MDP, and we therefore complement them by directly exploiting the tractability advantages presented by finite-state MDPs, such as closed-form return distribution moments and the possibility for exact inference on the environment dynamics, with methods leading to computationally efficient and accurate uncertainty representation.

## 3 BACKGROUND

**Dynamic Programming** A Markov Decision Process $\mathcal{M}$ (MDP) [Puterman, 2014] is given by a tuple $(\mathcal{S}, \mathcal{A}, r, P, \gamma, \rho)$, where $\mathcal{S}$ and $\mathcal{A}$ are the (assumed finite) state and action spaces respectively, $r : \mathcal{S} \to \mathbb{R}$ is the reward function, $P : \mathcal{S} \times \mathcal{A} \to \mathcal{P}(\mathcal{S})$ the transition kernel ($\mathcal{P}$ denoting a probability distribution over the corresponding set), $\gamma \in [0, 1)$ a discount factor and $\rho$ the distribution over initial states. Given a policy $\pi : \mathcal{S} \to \mathcal{P}(\mathcal{A})$, the *return* of an episode starting from state $s$ is a random variable given by $G^\pi(s) = \sum_{t=0}^{\infty} \gamma^t R_t$, where $R_t = r(s_t)$, $a_t \sim \pi(\cdot|s_t)$, $s_t \sim P(\cdot|s_{t-1}, a_{t-1})$ given that $s_0 = s$. For simplicity, we assume reward as known and only dependent on state. This is a natural modelling step for MDPs where a certain state is associated with a particular reward and in practice is common when constructing MDPs. Nonetheless our methods

extend naturally to more general formulations.

The expected value of $G$ is called the value function $V^\pi(s) = \mathbb{E}G^\pi(s)$, and it can be shown that with this definition, $V$ satisfies the Bellman equation

$$V^\pi(s) = r(s) + \gamma \sum_{a,s'} P(s'|s,a)\pi(a|s)V^\pi(s'). \quad (1)$$

Dynamic programming methods, such as value iteration, can evaluate $V$ and provide the policy that optimises $V$ [Sutton and Barto, 2018]. It can be shown that the value of any arbitrary policy $\pi$ is

$$\mathbf{v}(\pi) = (\mathbf{I} - \gamma\mathbf{T}(\pi))^{-1}\mathbf{r}, \quad (2)$$

with $\mathbf{v}$ and $\mathbf{r}$ being $|\mathcal{S}|$-dimensional vectors with $i^{\text{th}}$ element being $V^\pi(s_i)$ and $r(s_i)$ respectively (for $s$ the $i^{\text{th}}$ state in $\mathcal{S}$) and $\mathbf{T}(\pi)$ the policy-dependent transition matrix with element $i, j$ given by

$$\mathbf{T}_{i,j} = \sum_a \pi(a|s_i)P(s_j|s_i,a). \quad (3)$$

The term $(\mathbf{I} - \gamma\mathbf{T}(\pi))^{-1}$ can be interpreted as successor features, in terms of which the analytic solution for value has a simple form [Dayan, 1993]. For clarity we have highlighted here the dependence of $\mathbf{T}$ on $\pi$ and note that $\mathbf{r}$ does not depend on $\pi$ as we assumed state-dependent rewards.

**Return Distribution**    Unlike traditional distributional RL methods [Bellemare et al., 2017, 2023], we focus solely on the first two moments of the return distribution. This allows us to bypass the full distributional RL framework, as closed-form solutions for these moments can be obtained analytically for a given finite-state MDP.

Methods that solve the Bellman value equation (Eq. 1) can be extended to determine moments of the return distribution. For example, it can be shown that the variances of the return random variable $G^\pi(s)$ satisfy an analogous set of linear Bellman equations, with solution given in vector form by Sobel [1982]:

$$\mathbf{var}(\pi) = (\mathbf{I} - \gamma^2\mathbf{T}(\pi))^{-1}\mathbf{r}^{(\text{var})}(\pi), \quad (4)$$

where the vector of variances $\mathbf{var}$ has element $i$ corresponding to the variance at state $s_i$ and $\mathbf{r}^{(\text{var})}$ is the vector with element $i$ being

$$\mathbf{r}_i^{(\text{var})}(\pi) = \sum_j P^\pi(s_j|s_i)(r(s_i) + \gamma V^\pi(s_j))^2 - V^\pi(s_i)^2,$$

$$(5)$$

where $P^\pi(s'|s) = \sum_a \pi(a|s)P(s'|s,a)$.

**Bayesian Dynamics Model**    The dynamics model we employ is standard in Bayesian RL, and is equivalent to the one used in BAMDPs [Ghavamzadeh et al., 2015, Poupart et al., 2006] with an unchanging belief and similar to the one

proposed in Joshi et al. [2021], but stationary. By modelling the belief over the MDP's dynamics parameters, this line of work effectively captures the uncertainty due to not being able to fully narrow down the true underlying MDP: with a finite number of transitions, there may be several potential MDPs that could have generated the observations, to which we can assign posterior probabilities by using Bayes' rule. For our purposes, we take the reward function of the MDP as known (and deterministic), ultimately because in our applications we will define reward directly as a deterministic function of state, but treat the dynamics of the world as unknown.

Let $\theta_{s,a}^{s'}$ be a parameter representing the probability of transitioning to state $s'$ given action $a$ at state $s$, and consider a dataset of observed transitions $(s, a, r, s') \in \mathcal{D}$. The probability of transitioning to some next-state follows a multinomial distribution with parameters given by $\theta$, and we can specify a conjugate Dirichlet prior on these so that for each state-action the resulting posterior probability is also Dirichlet. Assuming a symmetric Dirichlet prior (independent across different state-actions) with parameter $\alpha_p$, the posterior distribution satisfies

$$p(\{\theta_{s,a}^{s_i}|s_i \in \mathcal{S}\}|\mathcal{D}) \propto \prod_j (\theta_{s,a}^{s_j})^{n_j + \alpha_p - 1}, \quad (6)$$

with $n_j$ being the number of times $s, a$ transitioned to state $s_j$ and the proportionality constant is given (in closed form) by the multivariate Beta function [Kotz et al., 2004].

When the number of possible outcomes, in this case next states, is large then inference on the Dirichlet parameters can be very data-inefficient: if a generic maximum-entropy prior parameter is employed it can assign a disproportionate amount of posterior probability to unobserved outcomes. To mitigate this, one may scale the prior parameter inversely to the number of outcomes, as done in a BAMDP context in Guez et al. [2012], or induce sparsity in the possible outcomes by modelling the belief of feasible next states through a hierarchical Bayesian model [Friedman and Singer, 1998]. We will address this same issue in section 5.3 by employing a sparse Dirichlet model.

**Aleatoric and Epistemic Uncertainty**    In order to quantify and distinguish between epistemic uncertainty due to ambiguity in MDPs $\mathcal{M}$ given limited data and aleatoric uncertainty in the return $G$, we use the common decomposition formula that arises after applying the law of total variance [Kendall and Gal, 2017, Joshi et al., 2021] to the return $G$:

$$\text{Var}\,G(s) = \underbrace{\text{Var}_\mathcal{M}\mathbb{E}\,G_\mathcal{M}(s)}_{\text{epistemic}} + \underbrace{\mathbb{E}_\mathcal{M}\text{Var}\,G_\mathcal{M}(s)}_{\text{aleatoric}}, \quad (7)$$

where we have made clear that the dependence on the return random variable $G$ is conditioned on the MDPs $\mathcal{M}$, so that the inner expectations and variances are marginalising over returns for a given MDP and the outer expectations and variances are marginalising over distributions of

MDPs. The epistemic variance term captures the overall variance in the expected returns due to ambiguity in the MDPs and the aleatoric variance term is an estimate of the intrinsic variance averaged over the posterior MDP distribution. Eqs. 2 and 4 allow us to determine $\mathbb{E}\, G_{\mathcal{M}}(s) = V_{\mathcal{M}}(s)$ and $\mathrm{Var}\, G_{\mathcal{M}}(s)$ exactly, while averages and variances over the MDPs can be approximated through Monte Carlo sampling of the posterior over MDPs. In the limit of infinite data, the epistemic variance will tend to 0 as the probability mass of the posterior focuses in on a specific $\mathcal{M}$, but the aleatoric term will not necessarily behave similarly.

**Bayesian Objective**  Beyond evaluating uncertainty, having a belief over the possible range of dynamics that an MDP can exhibit can allow us to account for this uncertain belief when carrying out control. Bayesian decision theory dictates that the optimal decision rule for a given prior belief and observed data is the one that maximises posterior expected value [Robert et al., 2007]. Thus, we seek to find a policy that maximises the posterior expected value objective

$$\max_{\pi} \sum_s \rho(s) \mathbb{E}_{\mathcal{M} \sim p(\cdot|\mathcal{D})} V_{\mathcal{M}}^{\pi}(s), \tag{8}$$

where the value of each state $\mathbb{E}_{\mathcal{M} \sim p(\cdot|\mathcal{D})} V_{\mathcal{M}}^{\pi}(s)$ has been marginalised with respect to the initial state distribution $\rho$. This approach is consistent with previous literature that establishes the benefits of optimising this objective for decision-making in uncertain MDPs [Nilim and Ghaoui, 2003, Iyengar, 2005, Xu and Mannor, 2006]. Thus, this objective will be one of the performance metrics we will use to evaluate different algorithms. Delage and Mannor [2010] addresses finding a policy that performs well on this objective, but their approach relies on a second-order expansion of the value in terms of the MDP parameters' posterior distribution moments, and must thus assume small posterior uncertainty to be successful.

## 4 METHODS

### 4.1 UNCERTAINTY QUANTIFICATION

Some proposed approaches for jointly estimating aleatoric and epistemic uncertainty in discrete-space MDPs either overlook uncertainty in the transition model [Festor et al., 2021] or rely on extensive Monte Carlo sampling [Joshi et al., 2021]. As a consequence, the former does not scale consistently with additional data (see Appendix D for empirical evidence for this claim) and we can improve on the latter in some regimes for the infinite-horizon MDP case by using closed-form expressions for the first two moments of the return distribution.

We present in Algorithm 1 a way to estimate posterior value, aleatoric and epistemic variances in Eq. 7, that exploits the finite-state stationary nature of the MDPs considered

here. Its computational complexity scales as $O(|\mathcal{S}|^3)$ due to requiring an $|\mathcal{S}| \times |\mathcal{S}|$ matrix inversion for each of the $N_M$ dynamics samples. In contrast, methods that rely on Monte Carlo return samples to estimate aleatoric and epistemic return will require a larger number of Dirichlet samples and large simulation trajectory lengths to achieve comparable accuracy, but no matrix inversion.

We investigate this trade-off quantitatively in Appendix A and conclude that the larger number of samples required for a full Monte Carlo-style evaluation (similar to Joshi et al. [2021]) is not worth the additional sampling overhead for the MDPs we are considering ($\gamma = 0.999, |\mathcal{S}| < 1000$). In particular, we show that for large $\gamma$, finding exact solutions for values using analytic forms will be more computationally efficient as longer rollouts become necessary to have accurate return samples and more posterior samples become necessary to decrease the error from Monte Carlo sampling. In principle one could also use some iterative policy evaluation scheme [Sutton and Barto, 2018] to solve for the first and second moments of the return distribution, sacrificing a small amount of accuracy but avoiding a matrix inverse calculation.

---

**Algorithm 1** Bayesian Value, Epistemic and Aleatoric Uncertainty Evaluation

---

**Require:** Policy $\pi$, state $s_i$, posterior distribution over transition parameters $p(\mathcal{M}|\mathcal{D})$

$\theta_{sa\{1:N_M\}}^{s'} \leftarrow N_M$ matrix samples from $p(\mathcal{M}|\mathcal{D})$

**for** $s \in S, s' \in S$ **do**

   $\{\mathbf{T}_{ss'}\}_{\{1:N_M\}} \leftarrow \sum_a \pi(a|s)\theta_{sa\{1:N_M\}}^{s'}$    $\triangleright N_M$ action-marginalised transition matrices

**end for**

**for** $t = 1$ to $N_M$ **do**

   $\mathbf{v}_t \leftarrow (\mathbf{I} - \gamma \mathbf{T}_t)^{-1}\mathbf{r}$    $\triangleright$ Eq. 2 for samples

   **for** $s_k \in \mathcal{S}$ **do**

      $V_k \leftarrow$ element $k$ of $\mathbf{v}_t$

   **end for**

   **for** $s_k \in \mathcal{S}$ **do**

      $\mathbf{r}_k^{(\mathrm{var})} \leftarrow \sum_j \{\mathbf{T}_{s_k s_j}\}_t (r(s_k) + \gamma V_j)^2 - V_k^2$

   **end for**

   $\mathbf{var}_t \leftarrow (\mathbf{I} - \gamma^2 \mathbf{T}_t)^{-1}\mathbf{r}^{(\mathrm{var})}$    $\triangleright$ Equation 4

   $v_t \leftarrow$ element $i$ of $\mathbf{v}_t$

   $var_t \leftarrow$ element $i$ of $\mathbf{var}_t$

**end for**

bayes_value $\leftarrow \frac{1}{N_M}\sum_{t=1}^{N_M} v_t$

aleatoric_var $\leftarrow \frac{1}{N_M}\sum_{t=1}^{N_M} var_t$

epistemic_var $\leftarrow \frac{1}{N_M-1}\sum_{t=1}^{N_M}(v_t - \text{bayes\_value})^2$

   **return** bayes_value, aleatoric_var, epistemic_var

---

## 4.2 POLICY IMPROVEMENT

Unlike with a single MDP, the objective in Eq. 8 does not always admit a deterministic optimal policy (we provide an example in Appendix B where the optimal policy is stochastic). For this reason, approaches analogous to classical dynamic programming cannot find an optimal policy. We suggest a gradient-based approach to optimise this objective in Algorithm 2. We approach the optimisation by taking stochastic gradient steps of the value objective with respect to a parametrised stochastic policy, which is made possible by the analytic form for value for given parameter samples. This approach is qualitatively distinct to the classic policy gradient in RL which estimates policy gradients from rolled-out trajectories [Sutton and Barto, 2018] . In contrast to other methods [Komorowski et al., 2018, Dimitrakakis, 2011] this does not introduce bias due to optimising only with respect to a finite number of transition samples: by re-sampling from the posterior every gradient step, we remove the bias that would occur by picking a smaller finite sample, and we note that all standard stochastic gradient optimisation guarantees regarding computational complexity or convergence to a local optimum will apply. For example, one can show that with appropriate learning rate scheduling, this convergence is guaranteed almost surely [Bottou, 1998] (although we empirically found that convergence was also achieved with a constant learning rate). Note that since $\gamma < 1$, all quantities (values, variances) are bounded, continuous and differentiable functions of policy parameters. We also remark that Algorithm 2 can be implemented faster computationally by reducing the batch size or by resampling the posterior periodically rather than at every gradient step (see Appendix C for computational benchmarks).

---

**Algorithm 2** Stochastic Gradient Policy Optimisation

---

**Require:** Initial deterministic $\pi$, posterior distribution over transition parameters $p(\mathcal{M}|\mathcal{D})$, initial policy softness parameter $\eta$, learning rate $\alpha$
  $\forall s \in \mathcal{S}, a \in \mathcal{A}, z_{sa} \leftarrow \log(\eta/(|\mathcal{A}| - 1))$
  $\forall s \in \mathcal{S}, z_{s\pi(s)} \leftarrow \log(1 - \eta)$    $\triangleright$ Set initial $\pi$ params
  **while** not converged **do**
      $\forall s \in \mathcal{S}, a \in \mathcal{A}$, let $\pi(a|s) \leftarrow \frac{\exp(z_{sa})}{\sum'_a \exp(z_{sa'})}$
      $\theta^{s'}_{sa\{1:n\}} \leftarrow n$ minibatch samples from $p(\mathcal{M}|\mathcal{D})$
      **for** $s \in S, s' \in S$ **do**
        $\{\mathbf{T}_{ss'}\}_{\{1:N_M\}} \leftarrow \sum_a \pi(a|s)\theta^{s'}_{sa\{1:N_M\}}$  $\triangleright N_M$
action-marginalised transition matrices
      **end for**
      $\mathbf{v}_{1:N_M} \leftarrow (\mathbf{I} - \gamma\mathbf{T}_{1:N_M})^{-1}\mathbf{r}$   $\triangleright$ Eq. 2 for samples
      $\mathcal{L} = -\sum_i \rho \cdot \mathbf{v_i}$  $\triangleright$ Posterior and state marginalised
      $\forall s \in \mathcal{S}, a \in \mathcal{A}, z_{sa} \leftarrow z_{sa} - \alpha\frac{\partial \mathcal{L}}{\partial z_{sa}}$  $\triangleright$ Gradient step
  **end while**
  $\forall s \in \mathcal{S}, a \in \mathcal{A}, \pi(a|s) \leftarrow \frac{\exp(z_{sa})}{\sum'_a \exp(z_{sa'})}$
    **return** $\pi$

---

## 5 EXPERIMENTS

Here we apply the proposed method on toy environments and a real-world clinical dataset. The toy environments demonstrate the salient features of our methods where ground-truth MDPs can be easily generated and interpreted, while the application to clinical data confirms its scalability to MDPs with practical use. We first examine uncertainty evaluation on interpretable gridworlds for a specific policy and then consider policy optimisation on gridworlds and synthetic MDPs. Finally we apply the same methods to the MIMIC-III dataset [Johnson et al., 2016], and present results on the impact that carrying out our approach has on this dataset's posterior expected value.

### 5.1 GRIDWORLD

We consider a gridworld with stochastic transitions: at each step there is a probability $p_{\text{rand}}$ of being pushed down regardless of action taken. Otherwise, the agent moves up, down, left or right by one square determined by the action. The observed transitions dataset $\mathcal{D}$ is generated by repeatedly spawning an agent in a non-terminal random state and carrying out a random action. Experiments are ran on the gridworld visualised in Fig. 1a. The results presented here are for datasets of varying sizes, where smaller ones are always subsets of any larger ones to ensure that the latter are strictly more informative.

**Uncertainty Quantification** We consider the policy uncertainty *evaluation* problem, comparing how results from our Bayesian approach differ from others when evaluating aleatoric and epistemic uncertainty for the policy that is optimal under the MLE dynamics parameter estimates. We see in Fig. 1b that the uncertainty quantification results from applying Algorithm 1 scale consistently with varying dataset size (epistemic uncertainty always becomes small with more data) and intrinsic stochasticity (higher $p_{\text{rand}}$ corresponds to higher aleatoric uncertainty). In contrast, we find that the approach in Festor et al. [2021] always leads to low epistemic uncertainty at the end of training, as the lack of knowledge of the underlying MDP is not modelled, and thus does not scale consistently with data. In Appendix D we visualise how epistemic uncertainty evolves during training with different datasets. We adapt their algorithm to carry out SARSA policy evaluation on the same, fixed policy and observe that it always tends to be small regardless of how informative the dataset is by the end of training. Additionally, as discussed in section 4.1, the computation of aleatoric and epistemic uncertainty through closed-form moments as in Algorithm 1 does not require averaging samples over episodic rollouts as in Joshi et al. [2021].

**Bayesian Policy Optimisation** An optimal memoryless policy that accounts for the model uncertainty maximises

the posterior expected value given in Eq. 8. We compare the performance on this objective of four policies: the optimal policy when transition probabilities are modelled as naive visitation frequencies (MLE-optimal policy), the optimal policy for the *expected* or marginalised (referred to as nominal) MDP (Nominal policy) [Dimitrakakis, 2011, Delage and Mannor, 2010], the policy derived from the second-order approximation of value in terms of posterior moments proposed in Delage and Mannor [2010] (Second order policy) and ours, described in Algorithm 2 (Gradient policy). In Algorithm 2 and the second order policy, we choose the initial policies to be a softened version of the Nominal policy ($\eta = 0.1$). Just like the MLE-optimal policy, the required amount of computation is one round of value iteration [Sutton and Barto, 2018] and is therefore a computationally negligible addition to the algorithm.

A further method that optimises a similar objective is the Multi-Sample Backwards Induction (MSBI policy) algorithm suggested in Dimitrakakis [2011]. However, this algorithm was originally devised to find a policy that achieves a near-optimal lower bound on the utility with respect to the Bayes-*adaptive* policy, which is a different task to the one considered here. Nonetheless, for completeness we report the corresponding results for this method in Appendix E, and found that it did not outperform ours on any of the metrics considered.

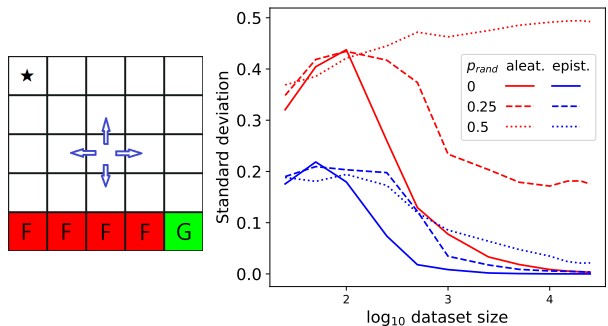

(a) Gridworld          (b) Aleatoric and epistemic uncertainty

Figure 1: Fig. 1a shows the gridworld used in the experiments. The terminal states are the failure **F** states (cliff) and the goal **G** state. The agent can move up, down, left, or right (or remain stationary if it hits the boundary of the grid). The transition dynamics have intrinsic stochasticity controlled by the probability $p_{\text{rand}}$, which is the probability of pushing the agent down regardless of action taken. Offline training datasets were generated by randomly sampling actions at random non-terminal states. State ★ is chosen as an exemplar state to plot state-dependent uncertainties. In Fig. 1b, the plot shows the epistemic (blue) and aleatoric (red) standard deviations as a function of training dataset size, with different levels of intrinsic stochasticity indicated by solid, dashed, and dotted lines.

We empirically find that the gradient-optimised policy consistently outperforms the other methods on optimising the posterior value objectives, especially in lower data regimes when optimising the Bayesian posterior value objective. Results from a sample run are presented in Fig. 2a for different dataset sizes. The corresponding relative performances of our method against both MLE-optimal and second order policies on the posterior value objective over 50 sets of generated datasets are presented in Fig. 2b, 2c and 2d with error bars (standard deviations), confirming that our method consistently outperforms the other two in terms of posterior value maximisation over a larger number of randomly-generated datasets.

## 5.2 SYNTHETIC MDPS

While gridworlds are convenient to interpret results relating to uncertainty disentanglement, they are not adequate for repeated experimentation and evaluation on multiple ground truth environments. Therefore, we present here results on unstructured, synthetic MDPs that allow us to meaningfully evaluate the ground-truth performance of learned policies on a large number of MDPs.

The MDPs we consider have 5 states, 5 actions and are generated by sampling the ground-truth transition probabilities independently for each state-action from a flat Dirichlet prior (with any state being a valid next state). To break the symmetry between states, we sample the state-dependent reward from a normal Gaussian and keep these constant throughout all experiments. The datasets are generated by sampling the outcome of visiting each state-action between 1 and 10 times, resulting in different dataset sizes. For each dataset size, we generate 250 different MDPs and datasets and train the various policies on these datasets. Finally, we roll out the policies and evaluate them on the ground-truth MDP that generated the data (for 1k steps and with $\eta = 0.5$). Similarly to the previous section, the intrinsic 'luck' associated with MDPs generated can affect the maximum value that each policy is able to achieve, so we focus on the difference in performance between methods for each MDP. In Fig. 3, we display the ground-truth relative performance of the various policies compared to ours. For all ground-truth results, we display standard error of the mean rather than standard deviation as we are interested in average performance across prior MDP samples rather than the variability over each individual sample. We also report the performance on the posterior expected value, as with the gridworlds, in Fig. 4. In Appendix F, we present the same results in Figs. 3 and 4 with the $y$-values rescaled to reflect fractional improvements rather than absolute values.

We notice that our stochastic gradient/based method consistently outperforms the others in the low data regimes, both for ground truth performance as well as optimisation performance. MLE and Nominal policies can be very sub-optimal

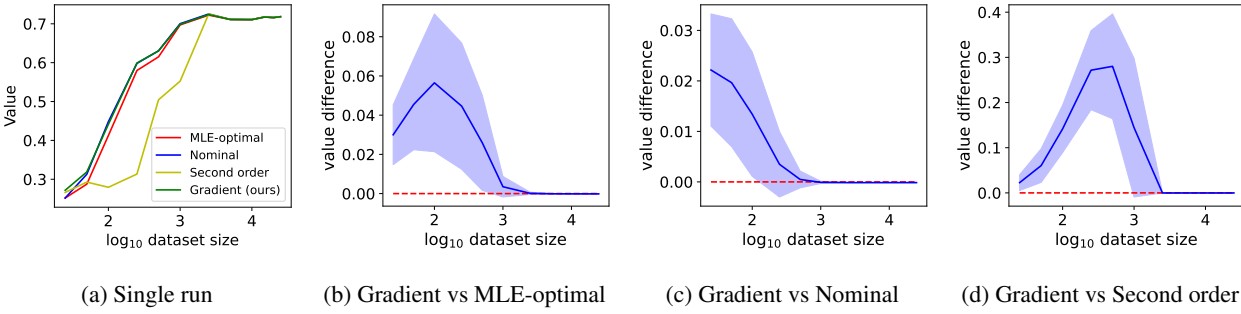

(a) Single run       (b) Gradient vs MLE-optimal       (c) Gradient vs Nominal       (d) Gradient vs Second order

Figure 2: Fig. 2a shows the average posterior expected return ('Value') as a function of dataset size for a single set of generated datasets, as in the objective in Eq. 8. The example gridworld has $p_{\mathrm{rand}} = 0.25$. As value will be dataset-dependent, we show the average and standard deviation between the pairwise difference in posterior values between ours and the other methods in Figs. 2b, 2c and 2d, where values above the red dashed line signify an improvement. These plots report the average and standard deviation across 50 generated datasets for each dataset size.

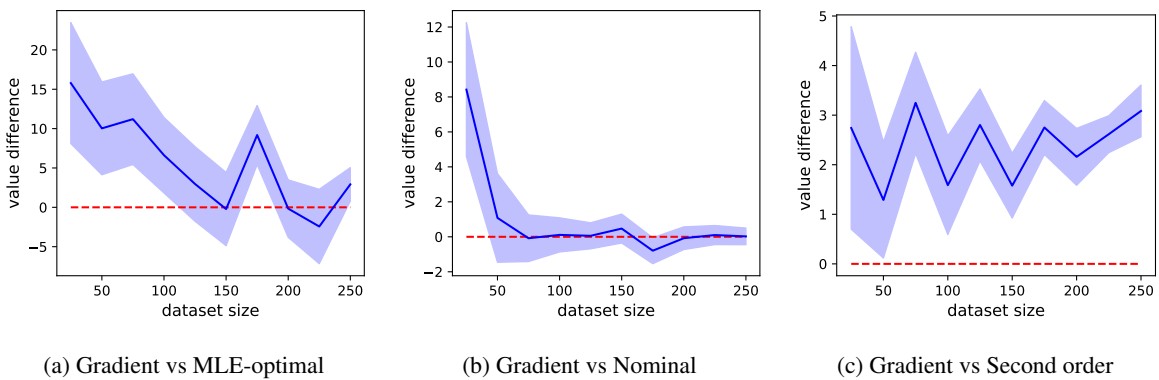

(a) Gradient vs MLE-optimal       (b) Gradient vs Nominal       (c) Gradient vs Second order

Figure 3: Ground truth pairwise difference in average performance (and shaded standard error of the mean) on the policies found by each method and rolled out on the ground-truth synthetic MDP. Regions above the red line correspond to improved performance with our method.

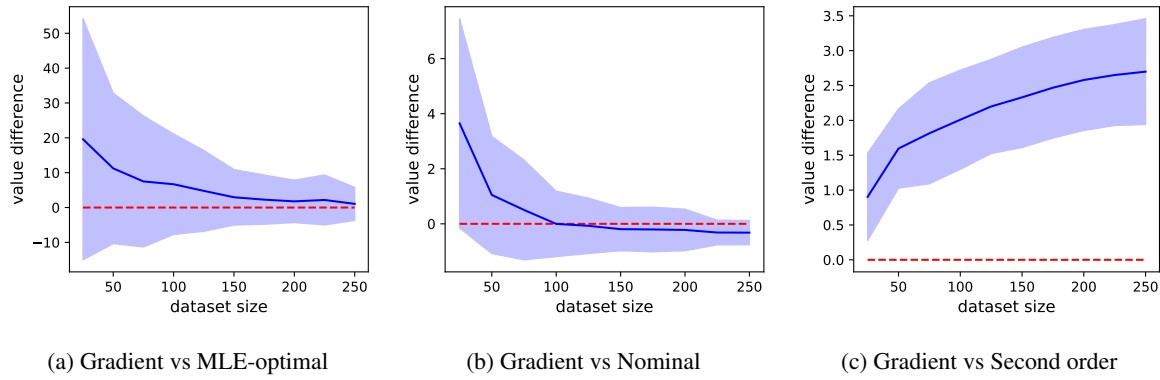

(a) Gradient vs MLE-optimal       (b) Gradient vs Nominal       (c) Gradient vs Second order

Figure 4: Average and standard deviation (shaded) of posterior expected value difference achieved by our method. Regions above the red line correspond to improved objective optimisation with our method.

in the low data regime and then significantly improve with more data, suggesting that with more data the gradient-based optimisation may not be necessary, whereas the second order policy is consistently slightly sub-optimal compared to ours.

## 5.3 CLINICAL DATA

We apply Algorithm 2 to the MIMIC-III dataset, as in Komorowski et al. [2018] and Festor et al. [2021], using the same clustering of 752 states and 25 actions. Two terminal states represent patient recovery and death, with reward 1 for a patient's recovery and 0 for death. Thus value corresponds to probability of survival when $\gamma \approx 1$ ($\gamma = 0.999$). As in Komorowski et al. [2018], actions at any state with fewer than 5 visits in the dataset are excluded. We address here two main points. First we confirm that our method can scale computationally to real-world MDPs and datasets. Secondly, we investigate the impact on the posterior expected value when employing our policy compared to the MLE-optimal one as in the original work.

Fig. 5 shows the posterior expected value of the two policies under two different choices of dynamics prior. Fig. 5a corresponds to a symmetric Dirichlet prior chosen via Bayesian model selection. The posterior probability mass over transition parameters still has a high entropy causing the agent to believe transitions are essentially random. In Fig. 5b, we employ a conservative sparse dynamics model that only includes the death state and any observed next states in the dataset as possible next-state outcomes for each state-action. Here, we notice that the posterior expected value can be significantly increased by using our policy optimisation algorithm, suggesting that we are in a data regime where the choice of algorithm for policy selection is important. We defer more detailed discussion and a visualisation of resulting uncertainties to Appendix G.

## 6 LIMITATIONS

Our methods are investigated for a specific category of Markov Decision Processes (MDPs) with finite states and known reward structures. While our method is well-suited for the lower data regimes, we empirically observe that it can be slightly suboptimal compared to the classical dynamic programming baselines, in particular the "Nominal" policy, in higher data regimes where uncertainty in the underlying transitions is low. Nonetheless, in practice this can be detected by comparing the Bayesian objective of the nominal policy to the posterior expected value achieved by the policy after our stochastic gradient optimisation and choose the one with the better posterior expected value. We have shown our approach can handle moderately-sized MDPs that carry practical real-world application possibilities in section 5.3). However, it relies on matrix inversion ($\mathcal{O}\left(|\mathcal{S}|^3\right)$ complex-

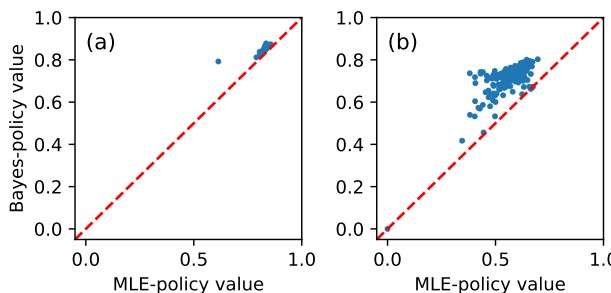

Figure 5: Posterior of each state (blue dots) under our policy and the MLE-optimal policy in the clinical MDP. Points above the diagonal indicate superior performance of our policy on the posterior expect value. The left plot (a) demonstrates the impact of policy choice on performance when employing Bayesian model selection with an optimal parameter of $\alpha_p = 0.072$. The right plot (b) shows the same result when using a prior selected through a conservative sparse dynamics model.

ity) so it cannot directly scale to much larger MDPs. In Appendix C we show empirical results on how our method scales to larger state-spaces on the computational side. One key limitation of our proposed methods towards real-world application is the sensitivity of the resulting policy and inferred values on the dynamics model prior used, especially when data is inadequate for effective inference across all dynamics priors. For example, we observe that the effects of having a sparse or evidence-optimised model can be significant on both the inferred policy and the associated posterior values (see Fig. 5) and exactly how to best include or combine these elements to select a prior that achieves consistently good performance on real-world MDPs is an important question and one that we defer to future work.

## 7 CONCLUSION

We have proposed methods to estimate Bayesian aleatoric and epistemic uncertainty in the outcome of finite-state space policies and to maximise posterior expected value. We offer a real-world example application of our method in a prominent case of discrete-state offline RL [Komorowski et al., 2018] in clinical decision support systems. In contrast to previous approaches that estimate such uncertainties in MDPs with finite states, we directly exploit the tractability of stationary MDPs to avoid potentially computationally expensive or inaccurate episodic rollouts (necessary in nonstationary MDPs [Joshi et al., 2021]) or employing ensemble of model-free approaches that may overlook dynamics uncertainty [Clements et al., 2019, Festor et al., 2021].

On the control side, we introduced a stochastic gradient-based method to optimise posterior expected value [Xu and Mannor, 2006] that, unlike previous approaches [Delage

and Mannor, 2010], does not make strong assumptions on the posterior's distribution and does not introduce bias from having a finite number of posterior samples [Dimitrakakis, 2011]. Through numerical simulations, we have shown that our method consistently improves on the posterior value objective as well as performance on ground-truth MDPs, particularly in low data regimes, when these are unknown and sampled from a given prior. Our method can be extended to optimise value over any distribution of MDPs that can be sampled from, including those with uncertain or more expressive rewards (of the form $R(s, a, s')$) as these also have differentiable closed-form expressions for value in terms of policy [Sobel, 1982]. We apply our method to a clinical dataset, confirming its computational scalability, and notice that the resulting policy significantly improves the posterior expected values compared to that in the original approach [Komorowski et al., 2018]. We suggested domain-specific conservatism in the dynamics model as a potential solution to new challenges that arise in this task and a starting point for further work towards finding offline policies with robust ground-truth performance in finite-state MDPs.

**Acknowledgements**

FV was supported by a Department of Computing PhD scholarship and AAF was supported by a UKRI Turing AI Fellowship (EP/V025449/1).

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

# A PROBABILISTIC EVALUATION BOUNDS

Here we provide a quantitative investigation into the choice of method to evaluate the quantities of interest for a given policy, including a comparison of the probabilistic bounds on the errors due to finite numbers of samples. We compare the efficiency required to achieve an evaluation within a certain accuracy $\varepsilon$ with a minimum probability $1 - \delta$ for methods that (i) carry out Monte Carlo sampling for every evaluation step and (ii) (ours, Algorithm 1) carry out an exact calculation of the return distribution moments and then Monte Carlo evaluation with samples from the dynamics posterior. The quantity we investigate in detail is the Bayesian value at a given state for a given policy (appearing in Eq. 8 for a given policy and state), and since aleatoric and epistemic uncertainties are calculated in very similar fashion, the conclusions regarding Bayesian value estimation will also carry through to the uncertainty quantification case.

## A.1 EXACT MOMENTS

The quantity of interest we wish to approximate is

$$\hat{V} = \mathbb{E}_{\mathcal{M}}(V_{\mathcal{M}}(s)), \tag{9}$$

where the expectation is taken over the Dirichlet posterior of MDP dynamics parameters. For a given set of dynamics parameters $\mathcal{M}$, we have access to the closed form expression for the first moment of the return distribution $V_{\mathcal{M}}(s)$ (in terms of policy, dynamics and reward) as presented in Eq. 2.

We assume a bounded reward $|r| \leq r_{\max}$ and employ the well-known form of the Hoeffding inequality Hoeffding [1994] valid for the random variable $S_n = \sum_{i=1}^{n} X_i$ with $X_i$ bounded and i.i.d. such that $\mathbb{E}(S_n) = \mu$:

$$\mathbb{P}(|S_n - \mu| \leq \epsilon) \geq 1 - 2 \exp\left(-\frac{2\epsilon^2}{n\Delta^2}\right) \tag{10}$$

with $\Delta$ being the size of the interval on which $X$ can take values.

In context, we take $X_i = \frac{1}{N_M} V_i$ as the closed-form expression for the value of the $i^{\text{th}}$ of the $N_M$ dynamics samples, so $\mu = \hat{V}$. From the boundedness assumption on the reward, we can also bound $|V_i| \leq \frac{r_{\max}}{1-\gamma} = V_{\max}$ and $\Delta \leq 2V_{\max}/N_M$. We require enough samples so that with probability at least $1 - \delta$ the error in our approximation of $\hat{V}$ is within $\epsilon$ of the true value. By the Hoeffding inequality, we can ensure this is the case by choosing $N_M$ such that

$$\delta \leq 2 \exp\left(-\frac{N_M \epsilon^2}{2V_{\max}^2}\right), \tag{11}$$

which corresponds to the smallest integer $N_M$ such that

$$N_M \geq \log\left(\frac{2}{\delta}\right)\left(\frac{2V_{\max}^2}{\epsilon^2}\right). \tag{12}$$

## A.2 MONTE-CARLO SAMPLING

The alternative method to using closed-form expressions for the moments of the return distribution given an MDP sample would be to in turn approximate these through Monte Carlo samples, as done in Joshi et al. [2021]. To do so, given the infinite horizon nature of the MDPs we are considering, we would have to accumulate rewards over a roll-out with a finite number of steps $T$, thus incurring in some error, which can be bounded above by $\gamma^T V_{\max}$. Note that the tightness of this bound will depend entirely on the reward structure of the MDP, and that this is not a source of error that can be reduced by repeatedly sampling transitions. For the purposes of the analysis presented, we will be generous in mostly ignoring the computational cost associated with sampling trajectories for a given MDP. In practice, sampling from a categorical distribution (i.e. sampling the trajectories for a given MDP) is significantly faster than sampling from a Dirichlet distribution (i.e. sampling the transition matrix), so we incorporate the overall computational cost of trajectory sampling into the modest condition that $T$ cannot be arbitrarily large, but assume infinite trajectory sampling capability otherwise. This assumption allows us to determine the value for the $i^{\text{th}}$ given MDP arbitrarily accurately up to this error, so that the distance between the true value $V_i$ to the accumulated finite sum of rewards $V_i'$ will be bounded by $|V_i - V_i'| \leq \gamma^T V_{\max}$.

Thus, we can consider the distance

$$\left| \hat{V} - \frac{1}{N_M} \sum_i V_i' \right| \leq \left| \hat{V} - \frac{1}{N_M} \sum_i V_i \right| + \left| \frac{1}{N_M} \sum_i V_i - \frac{1}{N_M} \sum_i V_i' \right| \tag{13}$$

$$\leq \left| \hat{V} - \frac{1}{N_M} \sum_i V_i \right| + \gamma^T V_{\max}, \tag{14}$$

so that if

$$\left| \hat{V} - \frac{1}{N_M} \sum_i V_i \right| + \gamma^T V_{\max} \leq \epsilon, \tag{15}$$

with probability at least $1 - \delta$, then the distance to the original estimate also satisfies

$$\left| \hat{V} - \frac{1}{N_M} \sum_i V_i' \right| \leq \epsilon. \tag{16}$$

with at least probability $1 - \delta$.

As such, we apply the Hoeffding inequality in the form

$$\mathbb{P}\left( \left| \hat{V} - \frac{1}{N_M} \sum_i V_i \right| \leq \epsilon - \gamma^T V_{\max} \right) \geq 1 - 2 \exp\left( -\frac{N_M^2 (\epsilon - \gamma^T V_{\max})^2}{2 V_{\max}^2} \right). \tag{17}$$

Note that this also imposes a minimum horizon truncation of $T > \log(\epsilon/V_{\max})/\log \gamma$. Explicitly including the probability threshold $\delta$ now corresponds to finding an $N_M$ such that

$$\delta \leq 2 \exp\left( -\frac{N_M^2 (\epsilon - \gamma^T V_{\max})^2}{2 V_{\max}^2} \right), \tag{18}$$

so

$$N_M \geq \log\left( \frac{2}{\delta} \right) \frac{2 V_{\max}^2}{(\epsilon - \gamma^T V_{\max})^2}. \tag{19}$$

This bound corresponds to a worsening by a factor of $(1 - \gamma^T V_{\max}/\varepsilon)^{-2}$ in the number of samples required to get comparable accuracy to the method that uses exact moments. For example, for the gridworld setup considered ($\gamma = 0.999$, $r_{\max} = 1$ and positing $\epsilon = 0.001$) would require an order of magnitude of $T \approx 10^5$ for every rolled out trajectory, (of which we are assuming to be able to carry out an arbitrarily large number to obtain this bound) at which point the contribution of the trajectory sampling to the bottleneck would be severe and require a completely different bound to take it into account. Thus, for the regime we consider, choosing to compute exact moments does save computation towards the computational bottleneck of taking samples from a Dirichlet posterior.

Note that aleatoric and epistemic uncertainty will behave similarly: aleatoric variance is an analogous expectation over the second instead of first moment (which we again can have in closed-form or can estimate through Monte Carlo samples) and the bound will be analogous. Similarly, for epistemic variance the error in return due to truncated trajectories will compound when calculating the variance over expected returns, and again we expect a similarly greater number of samples for $N_M$.

## B  STOCHASTIC OPTIMAL POLICY

Here we provide an illustrative example of how the Bayesian objective Eq. 8 for expected value when MDPs are sampled from some distribution may not have a deterministic optimal policy.

Consider the following 'casino' MDP with three states. State $s$ corresponds to the player being in the casino, where the possible actions are to play or leave. Being in state $s$ costs the player 1 unit of currency every time that state $s$ is visited. The outcome of leaving is to deterministically transition to a terminal state with no further rewards. On the other hand, playing has a stochastic outcome, with a probability $\theta$ of losing, in which case the player remains in state $s$, and a probability $1 - \theta$ of winning, in which case the player transitions to state $w$, where they receive a payout of $R$ units and then deterministically

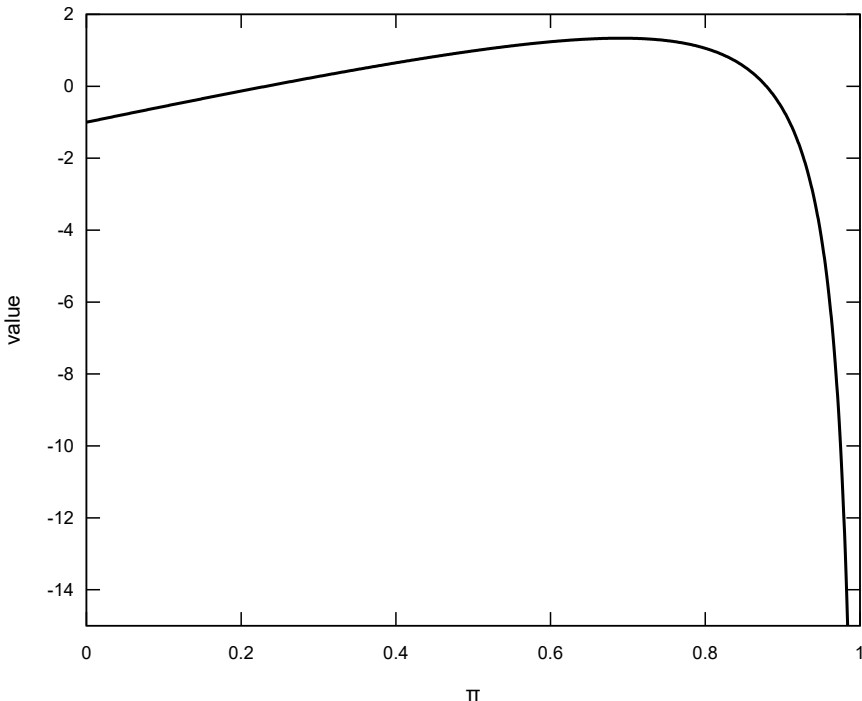

Figure 6: Plot of average value across a distribution of casino MDPs as a function of policy.

transition to the terminal state with no further rewards. Thus, each realisation of $\theta$ corresponds to a slightly different MDP with different probabilities of winning and therefore different optimal policies.

For a policy that plays with probability $\pi$ and leaves with probability $1 - \pi$, the Bellman equation for value of state $s$, $V$, under this policy is

$$V = -1 + \gamma(\pi\theta V + \pi(1 - \theta)R). \tag{20}$$

Solving for $V$ gives the value starting from state $s$ for a specific MDP:

$$V = \frac{-1 + \gamma\pi(1 - \theta)R}{1 - \gamma\pi\theta}. \tag{21}$$

We now consider the expected value when $\theta$ is sampled from some distribution. For example, if $\theta$ is sampled from a Bernoulli distribution with parameter $p = \frac{1}{2}$, the expected value of $\pi$ over this distribution of MDPs is

$$V = \frac{1}{2}\left(-1 + \gamma\pi R - \frac{1}{1 - \gamma\pi}\right). \tag{22}$$

We visualise this value as a function of $\pi$ in Fig. 6 for $R = 10$, $\gamma = 0.99$. The maximum value is not achieved at $\pi = 0$ or $\pi = 1$, but rather at $\pi \approx 0.69$. A policy that never plays achieves a value of 1, a policy that always plays a value of $-45.55$ and the optimal (stochastic) policy a value of about $1.34$. Thus, we have an example where there is no deterministic optimal policy.

## C   COMPUTATIONAL SCALABILITY

We present in Fig.7 empirical results with regards to scaling our method to larger state-spaces. The experiment we benchmark is the one on synthetic MPDs, as carried out in section 5.2 but with varying state-space sizes for two different posterior sample batch sizes. We also consider both the case where the posterior is resampled every gradient step (as in the synthetic MDP experiments) as well as the case where the posterior is only sampled once at the start of training. While the latter is not suggested in practice, the resulting copmutation time bounds the compute time that can be saved by resampling periodically between gradient steps rather than at every step during training.

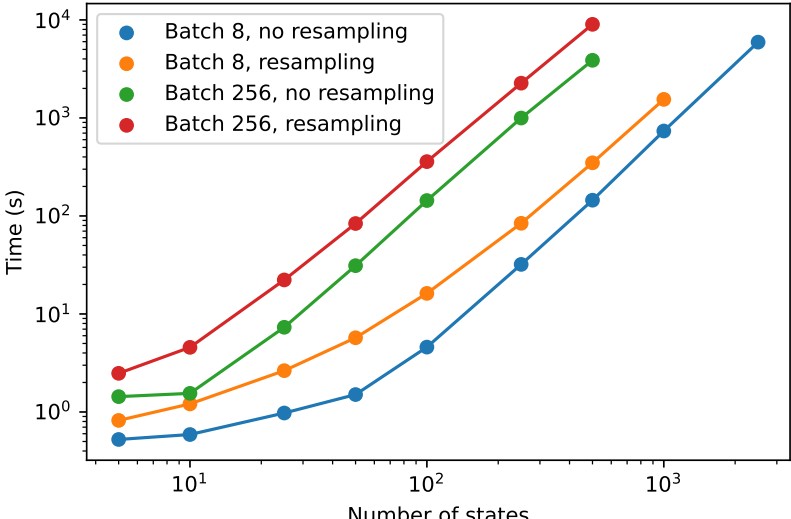

Figure 7: Computation time required to run Algorithm 2 for 1000 gradient steps. We display results for batches of sizes 8 and 256, for the case where the posterior is sampled before each gradient step (resampling) or only once at the start of training (no resampling).

## D   POLICY UNCERTAINTY EVALUATION

The policy we present and compare results for is the policy that optimises the maximum likelihood estimate (MLE) of the transition dynamics MDP, where transition probability is taken to be the relative frequency of observed transitions, which we refer to as the MLE-optimal policy.

Running SARSA policy evaluation on the methods proposed in Festor et al. [2021] explicitly shows that the epistemic uncertainty in the dynamics transition is not captured by the ensemble method used. Fig. 8 shows that with this setup, epistemic uncertainty correlates with loss but is independent of amount of data observed. This is visible as the curves collapse to small epistemic uncertainty values irrespective of data set size even though the amount of data in the smallest data set size (25) is smaller than the total number of transitions of the MDP (80). This is because it captures information on parametric training uncertainty but not of the dynamics model uncertainty.

## E   MULTI SAMPLE BACKWARD INDUCTION

We apply here a variant of the method MSBI presented in Dimitrakakis [2011], which outputs a policy that is near-optimal with respect to the Bayes-adaptive optimal policy under some assumptions. The main assumption is that the belief change between timesteps is bounded, and the authors show that a backwards induction greedy algorithm can yield a policy that has Bayesian utility which lower bounds the adaptive Bayes-optimal utility within an error term proportional to this bound. This optimal adaptive utility will, however, in general be different to the posterior value expectation that we are aiming to maximise with our policy, so the algorithm's purpose is not entirely aligned with ours. Nonetheless, in practice the proposed algorithm involves carrying out an analogous version of value iteration where at each timestep the iterative value is given by also marginalising over MDPs. This is also possible to implement and test in our setting so we provide here the relevant results for completeness, although we emphasise that the theoretical foundations and guarantees of near-optimality don't apply to our specific case. For a fairer comparison to the gradient-based methods, we also use the nominal policy as the starting policy in this algorithm.

**Gridworld**   The work suggests a number of posterior samples of the order of $(\epsilon(1 - \gamma))^{-3} \approx 10^{14}$ using $\gamma = 0.999$ and an error tolerance on the value of $\epsilon = 0.01$, which is a computationally intractable number of samples to store and process for transition matrices. Thus, we use a number of samples ($N_M = 32768$) and maximum number of iterations (2000) that roughly match the computation time of the gradient-optimised policy (30-60s depending on dataset without

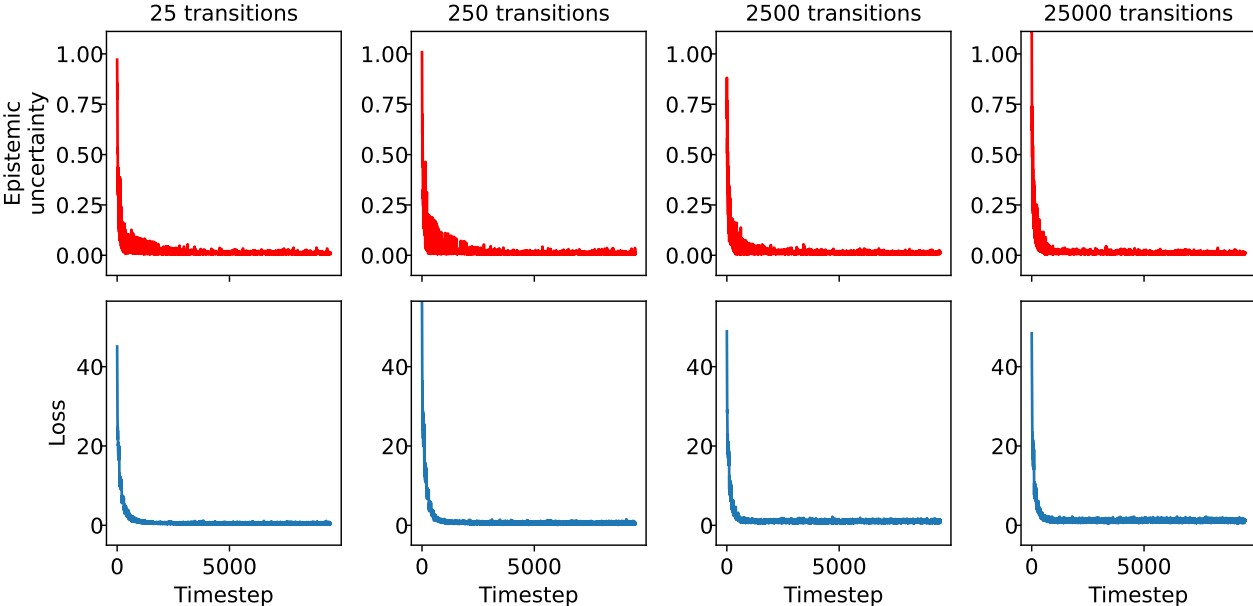

Figure 8: Plot of the epistemic uncertainty and loss as a function of training timestep demonstrating that epistemic is not accurately tracked by previous methods. Epistemic standard deviation (top row, red data) is quantified here over 10k time steps, corresponding to the agent carrying out transitions over many episodes. The corresponding ensemble quantile regression loss (bottom row, blue data) at each training timestep is shown below. Here we show as examplar the results for fixed policy using ensemble methods with a MLE-dynamics model for different number of observed transitions in the dataset generated by the gridworld with $p_{\text{rand}} = 0.5$. The value that the epistemic standard deviation converges to is always small for all visited states and independent of dataset size as the only notion of uncertainty captured in this setup is one of parametric uncertainty and not MDP uncertainty.

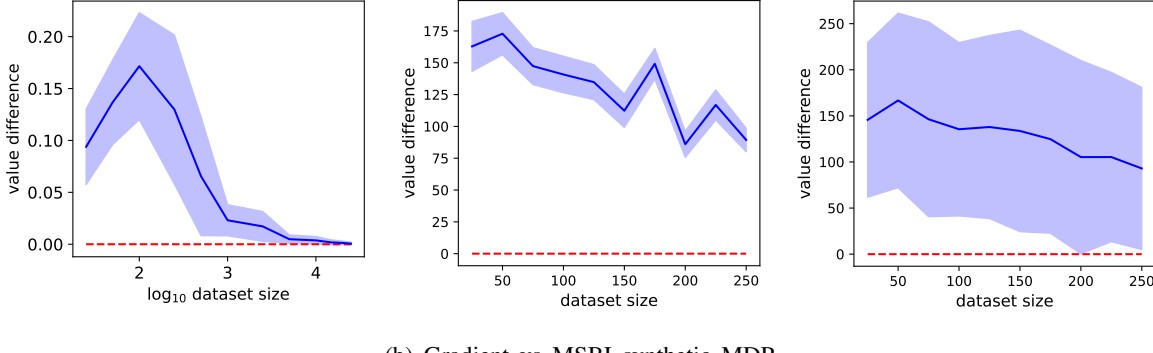

(b) Gradient vs MSBI synthetic MDP

(a) Gradient vs MSBI gridworld posterior ground truth performance (shaded stan- (c) Gradient vs MSBI synthetic MDP pos-
expected value                              dard error of the mean)                        terior expected value

Figure 9: Relative performance of MSBI on gridworld posterior expected value objective over 50 runs (Fig. 9a) and synthetic MDP ground truth performance (Fig. 9b, with shaded standard error of the mean) and posterior value objective (Fig. 9c) over 250 runs.

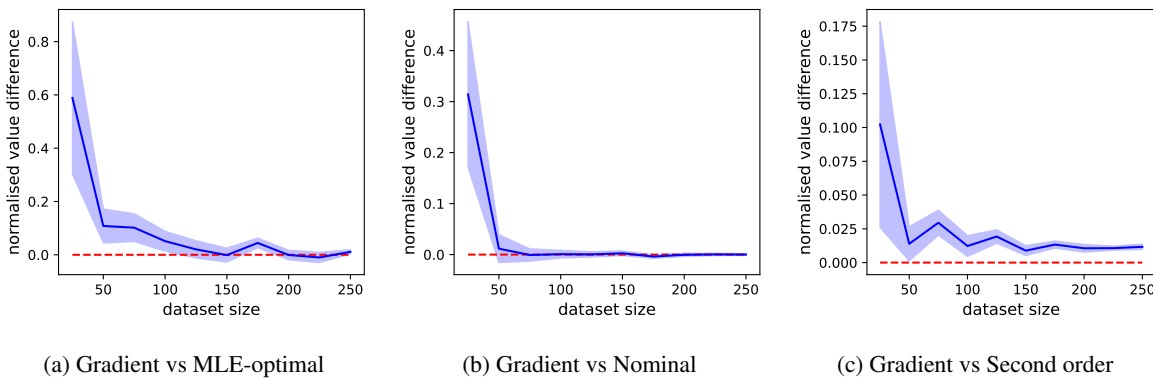

(a) Gradient vs MLE-optimal          (b) Gradient vs Nominal          (c) Gradient vs Second order

Figure 10: Ground truth pairwise difference in average performance (and shaded standard error of the mean) normalised by average performance of the Gradient (ours) method for each dataset. Regions above the red line correspond to improved performance with our method.

GPU acceleration for the gridworld experiments). In Fig. 9a we show the performance of MSBI on the relative posterior expected value objective for the same gridworld as in section 5.1 over 50 runs, where regions above the red line correspond to improved posterior value maximisation with our algorithm. As may be expected from the gap between the algorithm's original intention and our application, MSBI consistently underperforms with the exception of high data regimes, where the probability mass collapses on one MDP and the algorithm essentially reduces to value iteration.

**Synthetic MDPs**   We also apply MSBI to synthetic MDPs as presented in section 5.2 and report results in Figs.9b and 9c. Once again, due to the large number of experiments ran (250 runs each for each of the 10 different dataset sizes) we had to reduce the number of posterior samples to be $N_M = 2048$ and fix the maximum number of iterations to 500.

# F   SYNTHETIC MDPS RELATIVE PERFORMANCE

We display in Figs. 10 and 11 results corresponding to those presented in Figs. 3 and 4 but with the $y$-axis scaled by the value achieved by our method (resulting in a different scaling value for each dataset size), so that the new resulting plot can be interpreted as a fractional relative improvement.

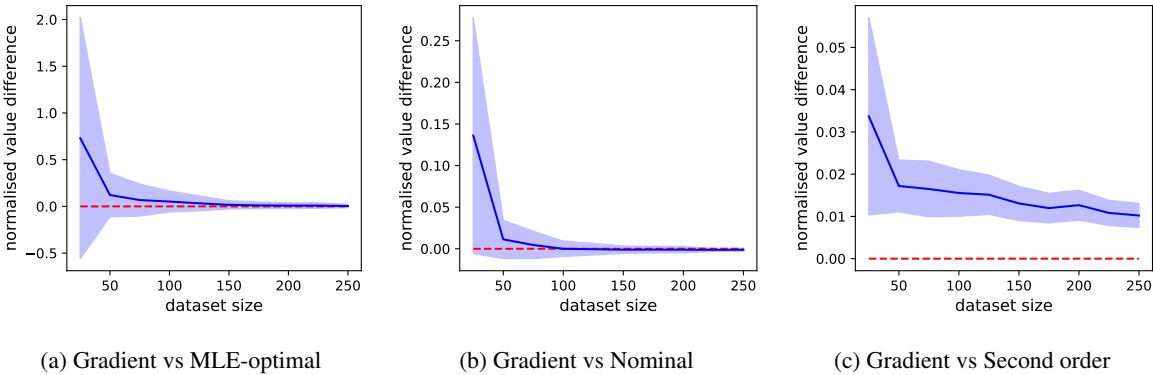

| (a) Gradient vs MLE-optimal | (b) Gradient vs Nominal | (c) Gradient vs Second order |

Figure 11: Average and standard deviation (shaded) of posterior expected value normalised by average performance of the Gradient (ours) method for each dataset. Regions above the red line correspond to improved performance with our method.

# G CLINICAL DATA DISCUSSION

## G.1 GENERAL DISCUSSION

Bayesian inference with Dirichlet distributions with a large number of possible outcomes (next states) is problematic, as mentioned in section 3 [Friedman and Singer, 1998], and careful thought must be given to what prior to employ. First we consider a Bayesian model selection approach: we assume all possible states are reachable and symmetric. This allows us to optimise the model evidence with respect to the unique parameter $\alpha_p$ of the prior, in the hope that specifying a prior which is more in line with the observations will lead to better inference (see Appendix G.2 for details). As expected, the optimal $\alpha_p$ is found to be much smaller than 1, $\alpha_p = 0.072$, giving less weight after inference to the prior than the maximum-entropy $\alpha_p = 1$ prior does. However, this approach still fails to accurately model our belief, which can be seen by considering the following scenario: suppose the patient is in a bad state and has two options, namely (a) try a treatment that has been attempted many times with rare success or (b) try a treatment that has always gone wrong, but has been tried a small number of times so has high uncertainty in the outcome. Option (b) is clearly not appealing, but the agent's posterior will still place significant probability mass on unobserved states in the presence of a small number of transitions, thus highly encouraging the agent to take the less visited action and assigning it a disproportionately high value. Upon inspection, this is exactly what is happening in the outlier state in Fig. 5a (at approximate coordinates $(0.6, 0.8)$), and the value given by this Bayesian posterior is likely unreasonable.

To address this, we introduce conservatism by considering only observed states and the death state as next possible states, thus ensuring a more conservative prior. Inducing conservatism in offline RL with datasets that do not adequately cover the full state-action space is in line with literature [Agarwal et al., 2020, Kumar et al., 2019], and conservative MDP models have found success in continuous offline RL by modulating reward [Yu et al., 2020, Kidambi et al., 2020] or dynamics [Guo et al., 2022], somewhat analogously to what is being proposed here. By only including observed or negative outcomes, the agent is unable to place probability mass on unsupported next-states and therefore use high uncertainty to inflate the value of poorly visited actions in bad states. The scarcity of outcomes allows for meaningful inference using a maximum-entropy prior with $\alpha_p = 1$, and a high-entropy prior is favorable from a conservatism standpoint. It encourages the agent to select actions that have sufficient support to offset the high prior probability mass assigned to the death state. The Bayesian values inferred with this setup are presented in Fig. 5b. Fig. 5 shows the possible improvement, according to the Bayesian posterior value, of employing the Bayesian gradient-optimised policy compared to the MLE-optimal policy used in Komorowski et al. [2018], resulting in higher probability of survival (according to the dynamics model). In particular, we note that employing the gradient-optimised policy improves the value, and therefore corresponding approximate probability of survival, by about $2.1\%$ when averaged across states, with a maximum improvement on a particular state of $17.8\%$, according to the conservative Bayesian dynamics model.

In Fig. 12 we show how the MIMIC-III states aleatoric and epistemic uncertainties are related. The values are computed using the same conservative dynamics model of Fig. 5b.

As expected for the particular reward structure of the MDP considered, aleatoric uncertainty and average Bayesian value are strongly related: since the return variable is approximately binomial (approximately 1 for success and 0 for failure) its mean

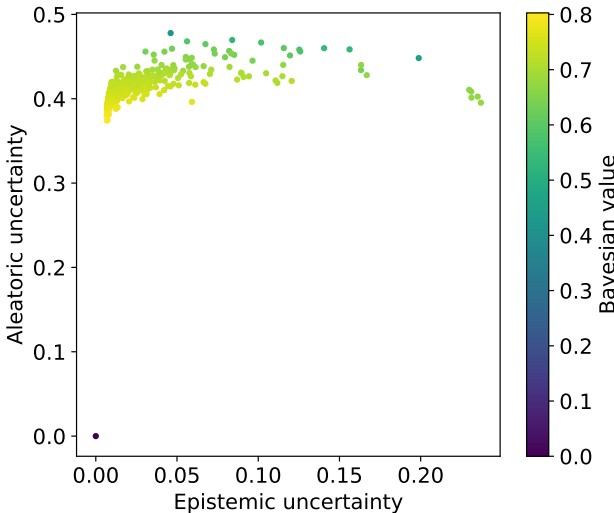

Figure 12: States plotted according to their epistemic and aleatoric standard deviations. Each dot represents a state, with its colour corresponding to its average value according to the Bayesian posterior.

and variance are related straightforwardly. Note this will not be true for MDPs with more general reward structures.

## G.2 BAYESIAN MODEL SELECTION

To determine the prior that for the dynamics model with results presented in Fig. 5a, we carry out Bayesian model selection by minimising the negative log-marginal likelihood of the data with respect to the parameter $\alpha_p$. To remain consistent with the limitation that only actions observed at least 5 times in the data should be employed at each state, we only use the data for such state-action transitions when determining the optimal $\alpha_p$.

For each state-action, the full form of the Dirichlet prior in terms of $\alpha_p$ is Friedman and Singer [1998]

$$p(\{\theta_{s,a}^{s_j}|s_i \in \mathcal{S}\}) = \frac{\Gamma(|\mathcal{S}|\alpha_p)}{\Gamma(\alpha_p)^{|\mathcal{S}|}} \prod_j (\theta_{s,a}^{s_j})^{\alpha_p-1}, \tag{23}$$

where $\Gamma$ is the gamma function. The likelihood is

$$p(\mathcal{D}|\theta) = \prod_j (\theta_{s,a}^{s_j})^{n_j}, \tag{24}$$

with $n_j$ being the number of observed transitions from state-action $s, a$ to state $s_j$. Hence, the model evidence is

$$p(\mathcal{D}) = \int d\theta p(\mathcal{D}|\theta)p(\theta) \tag{25}$$

$$= \frac{\Gamma(|\mathcal{S}|\alpha_p)}{\Gamma(\alpha_p)^{|\mathcal{S}|}} \frac{\prod_j \Gamma(\alpha_p + n_j)^{|\mathcal{S}|}}{\Gamma(|\mathcal{S}|\alpha_p + N_{s,a})}, \tag{26}$$

with $N_{s,a}$ being the number of observed transitions from state-action $s, a$. Since transitions are independent across state-actions, taking the negative logarithm of this quantity and summing across all state-actions results in the overall negative log-marginal likelihood for the dataset in terms of $\alpha_p$. The resulting function of $\alpha_p$ is visualised in Fig.13 and attains a minimum value at approximately $\alpha_p = 0.072$.

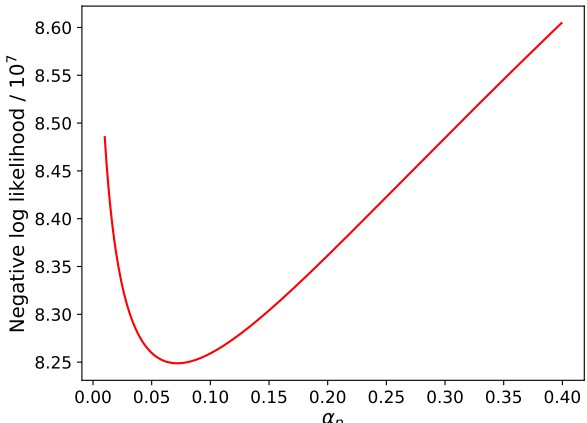

Figure 13: Negative log-marginal likelihood for clinical data dynamics model against parameter $\alpha_p$ of the prior.