# OpenReview forum: "Offline Bayesian Aleatoric and Epistemic Uncertainty Quantification and Posterior Value Optimisation in Finite-State MDPs"
_auai.org/UAI/2024/Conference — UAI 2024 poster_

### Official Review · Reviewer_nXGx · 2024-03-07

**Q2-1 Originality-Novelty:** 3
**Q2-2 Correctness-Technical Quality:** 3
**Q2-5 Clarity Of Writing:** 3

**Q1 Summary And Contributions:**

The authors provide a method to quantify and disentangle different types of uncertainty in MDPs. They also give a novel approach to optimize Bayesian posterior expected value which does not rely on strong assumptions about the MDPs’ posterior.

**Q2-3 Extent To Which Claims Are Supported By Evidence:**

3: Good: the main claims are supported by convincing evidence (in the form of adequate experimental evaluation, proofs, (pseudo-)code, references, assumptions).

**Q2-4 Reproducibility:**

3: Good: key resources (e.g. proofs, code, data) are available and key details (e.g. proofs, experimental setup) are sufficiently well-described for competent researchers to confidently reproduce the main results.

**Q3 Main Strengths:**

The method introduced seems new and compelling. The paper is well-written.

**Q4 Main Weakness:**

The theory could be developed further. This may result in an improvement in downstream task performances.

**Q5 Detailed Comments To The Authors:**

I enjoyed reading the paper, and this is reflected in the scores. The theory around the proposed methodology, though, could be further improved in the future. Allow me to suggest something in that direction. As shown in [1], ensembles suffer from shortcomings in quantifying and disentangling between different types of uncertainties. In view of this, in future work, the authors could explore a credal set approach to uncertainty quantification [2,3]. In [3], the authors develop an active learning methodology based on a loss incorporating credal sets. Maybe this could be of interest for the authors.

---

[1] https://arxiv.org/abs/2302.00704
[2] https://arxiv.org/abs/2302.09656
[3] https://arxiv.org/abs/2308.14815

**Q9 Complying With Reviewing Instructions:**

Yes

---

> ### Author Rebuttal · Authors · 2024-04-09
>
> We thank the reviewer for the comments on our work and appreciate the referenced literature, which is a relevant direction for future work as an alternative to the particular Bayesian view we are investigating in our manuscript.

---

### Official Review · Reviewer_KFZD · 2024-03-17

**Q2-1 Originality-Novelty:** 2
**Q2-2 Correctness-Technical Quality:** 3
**Q2-5 Clarity Of Writing:** 3

**Q1 Summary And Contributions:**

The manuscript develops a method to quantify Bayesian uncertainty while accounting for it in non-adaptive decision-making from static datasets in finite-state Markov Decision Processes whose dynamics are not given. In particular, the author/s introduce/s a method for recovering epistemic and aleatoric uncertainty together with a new approach for optimising Bayesian posterior expected value which avoiding to make strong assumptions about the posterior of the Markov Decision Processes. Uncertainty quantification and the performance of the proposed algorithm for Bayesian posterior optimisation is shown by considering elementary and interpretable gridworlds which allow ground-truth evaluations on synthetic models. The paper also show that the proposed approach can scale in terms of computational time by applying it to a clinical decision support system which allow clinicians ot make real-time recommendations for sepsis treatment in intensive care units.

**Q2-3 Extent To Which Claims Are Supported By Evidence:**

2: Fair: the main claims are somewhat supported by evidence (but the experimental evaluation may be weak, or does not match entirely with the claims, important baselines may be missing, proofs contain important ideas but lack rigor, algorithmic details are only discussed superficially, references are imprecise, assumptions are not sufficiently motivated or explicated, etc.).

**Q2-4 Reproducibility:**

3: Good: key resources (e.g. proofs, code, data) are available and key details (e.g. proofs, experimental setup) are sufficiently well-described for competent researchers to confidently reproduce the main results.

**Q3 Main Strengths:**

1) the tackled problem is extremely relevant both in terms of theory and practice
2) the proposed approach is sound
3) the proposed approach outperforms state of the art approaches in low data regimes
4) the epistemic uncertainty always becomes small with more data while this is not true for other methods
5) the mathematical treatment is well structured and described

**Q4 Main Weakness:**

1) the proposed approach tends to be outperformed by state of the art approaches in mid to high data regimes
2) the complexity of the proposed approach makes it difficult to scale to significant real world applications
3) the synthetic data example is too simple in my humble opinion, too few states and too few actions

**Q5 Detailed Comments To The Authors:**

I enjoyed reading the paper, it is well structured and written, while tackling a relevant problem when sequential decision making has to be made by taking into account the uncertainty in all the two basic components.
I would have appreciated whether the synthetic data example was complemented by other synthetica data where more complex problems are analyzed. Furthernore, I would like the paper to drill more on the fact that the proposed approach seems to work on low data regimes while it suffers when the amount of data increases. I also found that the real world example on sepsis could have been betetr described and analyzed.

**Q9 Complying With Reviewing Instructions:**

Yes

---

> ### Author Rebuttal · Authors · 2024-04-09
>
> We appreciate the thorough review and thoughtful comments directed towards our work and are pleased that the reviewer agrees with the relevance of the problem tackled and the soundness of our approach. We would like to address here some of the points raised.
>
> Weakenss 1:
> In the high data regime with small uncertainty, we do expect methods that use classical dynamic programming approaches (as in some of the baselines) to slightly outperform ours as they do not compromise with having to search over all stochastic policies or other issues with convergence or the possibility of running into local optima. Nonetheless, we note that this can be detected by comparing the Bayesian objective of the nominal policy to the objective achieved by the policy after our stochastic gradient optimisation and choose the one with better objective. Thus, in practice it would be possible to implement an approach that achieves the best of both. We entirely agree that our method is particularly well suited, by design, for MDPs with high uncertainty and lower amount of data and we specifically mention the observed empirical benefits in the lower data regime in sections 5.1, 5.2 and the conclusion. Nonetheless, we also agree that further emphasis on this point is beneficial to the presentation of the paper and have therefore added text in the introduction (“We empirically demonstrate its performance and scalability, providing results on gridworlds and synthetic MDPs with varying offline dataset sizes, where we observe benefits in MDPs with higher uncertainty and lower data.”), and in the limitations section (“While our method is well-suited for the lower data regimes, we empirically observe that it can be slightly suboptimal compared to the classical dynamic programming baselines, in particular the “Nominal” policy, in higher data regimes where uncertainty in the underlying transitions is low. Nonetheless, in practice this can be detected by comparing the Bayesian objective of the nominal policy to the posterior expected value achieved by the policy after our stochastic gradient optimisation and choose the one with the better posterior expected value.”).
>
> Weakness 2:
> We would like to note that the MDP setup for the clinical example used is identical to that of previous work [1] with an MDP of the same state-space size. We are able therefore to computationally scale up our method to significant real world applications.
>
> Weakness 3:
> We agree that the synthetic toy example is significantly more simple than an MDP that would be useful in practice; the focus of this experiment was to empirically demonstrate the fundamental soundness of the algorithm and to do so we prioritised running a large number of experiments (250 training runs for 10 different dataset sizes) to get a large sample size rather than investing computation time into larger MDPs.
>
> [1] Komorowski, M., Celi, L.A., Badawi, O. et al. The Artificial Intelligence Clinician learns optimal treatment strategies for sepsis in intensive care. Nat Med 24, 1716–1720 (2018). https://doi.org/10.1038/s41591-018-0213-5

---

### Official Review · Reviewer_4jFt · 2024-03-19

**Q2-1 Originality-Novelty:** 3
**Q2-2 Correctness-Technical Quality:** 3
**Q2-5 Clarity Of Writing:** 4

**Q10 Ethical Concerns:**

No.

**Q1 Summary And Contributions:**

The paper develops a method for separating epistemic and aleatory uncertainty in finite state Markov decision processes, from a Bayesian perspective. This has potential use in decision making, where decision makers need to know how much uncertainty in the posterior can be attributed to prior epistemic assessments. The method is numerically tested, validated, and compared.

**Q2-3 Extent To Which Claims Are Supported By Evidence:**

3: Good: the main claims are supported by convincing evidence (in the form of adequate experimental evaluation, proofs, (pseudo-)code, references, assumptions).

**Q2-4 Reproducibility:**

4: Excellent: key resources (e.g. proofs, code, data) are available and key details (e.g. proof sketches, experimental setup) are comprehensively described for competent researchers to confidently and easily reproduce the main results.

**Q3 Main Strengths:**

The paper is excellently written, with practically no errors in grammar. The problem and objectives of the paper are explained clearly, and are comprehensible to readers not immediately in the field. The algorithm by itself is interesting and has applications that extend well beyond the specific problems studied in the paper.

**Q4 Main Weakness:**

I have some concerns (noting that I am by no means an expert on the topic of Bayesian modelling of MDPs, so it may well be that I have missed something):
* The choice of prior, which represents the initial epistemic uncertainty (as far as I understand), receives little attention. Why choose a symmetric Dirichlet - is this purely for computational reasons?
* The authors argue in Section 5 that the prior needs to be informed from the data, by only assigning mass to state transitions that occur in the data. In a "pure" Bayesian setting however one should not use the data to inform the prior, and this will somewhat conflate epistemic and aleatory uncertainty.
* The application has an alpha value that is very close to zero, meaning that the data drives most of the posterior. However, it is unclear to me from the paper how this impacted the separation of epistemic and aleatory uncertainty in the posterior, from the given plots.

**Q5 Detailed Comments To The Authors:**

Introduction: To improve the reader's understanding of the importance of the results derived, it would be helpful to add a brief discussion of why the separation between aleatory and epistemic uncertainty is important, and how it may affect decision making

p4, algorithm 1:

* What is "i" in "element i of \vec{v}_t" and "element i of \vec{var}_t"? Is it the state we are interested in (from s_i mentioned at the top)? I wasn't clear as to why this step is necessary, since the equations for bayes_value etc. could be interpreted in a pointwise manner, and the algorithm calculates the values across all states simultaneously anyway.

* Somewhat related, why is V_k introduced? Can't we simply use \vec{v}_t?

p8 right col, last par: citation of Sobel needs brackets

**Q9 Complying With Reviewing Instructions:**

Yes

---

> ### Author Rebuttal · Authors · 2024-04-09
>
> We appreciate the thorough review of our work and the constructive comments for improvement of our manuscript.
>
> *Main weaknesses*
>
> Yes, the computational convenience that arises from Dirichlet being conjugate to the multinomial distribution makes it standard in the MDP literature as choice of prior. Note that our methods also straightforwardly apply to any other prior provided that the posterior distribution can be sampled from.
>
> We understand the view that in a pure Bayesian setting one would prefer not use data to inform prior belief. However, in practice accurately encoding belief in a prior that allows for computationally convenient inference (such as a conjugate prior) is very difficult. Thus, we resort to using some aspects of the data when devising our model and prior, which is also common in Bayesian model selection. While the choice of prior does impact the quantification and disentanglement of uncertainty, we did not investigate this particular aspect in detail in our work.
>
> *Detailed comments*
>
> This is a correct interpretation, it is true that the values are all calculated in parallel and this is how the algorithm is in fact implemented in code. We chose this notation for pseudocode as it was the clearest and most streamlined way we could convey the calculations carried out that was consistent with the notation in the rest of the manuscript.

---

### Official Review · Reviewer_22NV · 2024-03-23

**Q2-1 Originality-Novelty:** 2
**Q2-2 Correctness-Technical Quality:** 3
**Q2-5 Clarity Of Writing:** 4

**Q1 Summary And Contributions:**

The paper studies an offline decision making setting for finite state MDPs where limited data is available, where it is important to account for uncertainty in the dynamics when learning a policy.  The paper proposes an algorithm for evaluating aleatoric and epistemic uncertainty in the value function, which uses closed-form expressions for expected value and value uncertainty (available for finite-state MDPs), and a related algorithm for policy optimization, which is compared to baseline algorithms on gridworlds, synthetic MDPs, and a healthcare dataset for offline RL.

**Q2-3 Extent To Which Claims Are Supported By Evidence:**

2: Fair: the main claims are somewhat supported by evidence (but the experimental evaluation may be weak, or does not match entirely with the claims, important baselines may be missing, proofs contain important ideas but lack rigor, algorithmic details are only discussed superficially, references are imprecise, assumptions are not sufficiently motivated or explicated, etc.).

**Q2-4 Reproducibility:**

4: Excellent: key resources (e.g. proofs, code, data) are available and key details (e.g. proof sketches, experimental setup) are comprehensively described for competent researchers to confidently and easily reproduce the main results.

**Q3 Main Strengths:**

The paper is well written, and considers an important problem with real-world relevance in safety critical or high-stakes settings.  I think the approach of focusing on medium-sized MDPs where a closed-form solution can be exploited is worth exploring.

Overall, the experiments (including a real-world healthcare dataset for offline RL) provide some evidence that the proposed algorithm is more effective than alternatives in the low-data regime.

**Q4 Main Weakness:**

The algorithms are relatively straightforward; I think the novelty/innovation is incremental.

The paper does not include theoretical analysis of the proposed algorithms.

I also felt it was not entirely clear how significant vs incremental the empirical improvement over the baseline methods was, or how robust the improvement is to the choice of prior.

See Q5 for more detailed comments/concerns.

**Q5 Detailed Comments To The Authors:**

In the experiments section (e.g. Figure 3), it wasn’t quite clear how significant the value difference compared to baseline methods was.  In Figure 2, the proposed method only slightly outperforms two of the baselines in terms of value (the value difference is small compared to the absolute value).  In Figure 3 (and 4) since only the difference is shown, it’s hard to tell if this is a significant fraction of the absolute value.  Also, the fact that the plots in Figures 2-4 each scale the y-axis differently make it harder to visually compare the different baselines. (For instance, the value difference with the ‘Nominal’ baseline is consistently smaller.)

It also looks like the baselines are from papers from 10+ years ago – can you comment on why more recent works don’t provide stronger baselines?

For the clinical data experiment, the proposed method shows improvement (Fig. 5b) when using a conservative prior, but less so when using a symmetric Dirichlet prior.  How do we know that this improvement is not just an artifact of choosing a prior that works better?  Or why is this a practically realistic prior, and not overly conservative?

I’d also be curious to see more discussion of why some real-world MDPs will be large enough to be challenging, but have a small enough state space and large enough time horizon (gamma)  for the proposed approach to have an advantage.  I think clarifying this could make the paper more practically compelling.  On this note, I think it would also be good to see some empirical comparison of computational cost relative to baselines.

Typo in Figure 5 caption: “expect value”

**Q9 Complying With Reviewing Instructions:**

Yes

---

> ### Author Rebuttal · Authors · 2024-04-09
>
> We thank the reviewer for the appreciation of the relevance of the task we tackle, the thorough examination of our manuscript and for the constructive criticism of our work. We address here some of the concerns and questions raised.
>
> *Main weaknesses*
>
> While we agree that the methods we provide are fairly simple solutions, we disagree that the novelty is incremental. On the policy improvement side, our algorithm is unique. It provides significant novelty in that it finds an optimal non-adaptive policy without making strong assumptions about the MDP distribution the objective is optimised over (for example [1] assumes small higher-order distribution moments, which we empirically demonstrate in our experiments does not perform well in medium to lower-data regimes). We are able to do so by using a closed-form expression for value (from theory) to cast the problem of finding an optimal policy as a stochastic gradient optimisation problem. The novelty, therefore, lies in casting the problem in an appropriate form that allows us to solve the specific task we consider rather than in complex algorithmic contributions. We stress that in doing so we are able to find an algorithm that searches for an optimal policy over the space of stochastic policies (as the optimal policy here may be stochastic, as described in in Appendix B) which does not fundamentally rely on strong assumptions about the MDP posterior distribution, only requiring the ability to sample from it.
>
> We at puzzled by the statement claiming an absence of theoretical analysis, we argue that we do in fact present an analysis for the methods we suggest which provides justification for our methods. On the policy evaluation side, Appendix A provides a quantitative discussion on the computational tradeoff associated with using closed-form expressions. On the other hand, the policy improvement is implemented as a stochastic gradient algorithm, and we describe how all theoretical properties of standard stochastic gradient algorithms apply to our formulation. The brief discussion of the main consequences of this is provided at the end of Section 4.
>
> We appreciate that from the results provided for the synthetic data only show absolute value difference, making it difficult to judge the relative gain in performance. Thus, we are including complementary figures (https://figshare.com/s/dbfb0b3a6b092cd0e43a with file name matching representing the normalised version of the corresponding figure in the manuscript) with the corresponding normalised (fractional) value differences, where the plotted y-values are the original value difference divided by the mean absolute ground truth (fig 3) or bayesian posterior expected value (fig 4) achieved by our stochastic gradient method on each dataset (resulting in one scaling constant per dataset size). In this way, the plotted values can be interpreted as the relative fractional improvement of our method compared to the baselines. The plots show an improvement of at least 10% gain on the ground-truth performance in the lowest data regime from our method across the baselines (figure 3) .
>
> We considered reporting figures with the same y-axis limits but, as the reviewer correctly points out, different methods consistently achieve significantly different performances, and we found that having figures with separately scaled axes was the clearest way to present our results.
>
> Much of the recent research efforts into sequential decision making from fixed datasets have been focused on continuous state-action spaces with function approximators. Even in finite state-spaces, other formulations such as finding optimal adaptive or worst-case robust policies have received more attention and the task of finding a non-adaptive Bayesian-optimal policy has remained understudied. Nonetheless, we think that such a problem formulation is a very natural one to tackle, and we have highlighted the works that are relevant to a Bayesian decision-theoretic framing of the problem in finite state spaces and have included these in our baselines. With our work, we are providing a method that addresses the open question of finding a Bayes-optimal policy without requiring strong assumptions about the distribution of MDPs that is being optimised over.

---

### Official Review · Reviewer_S8Nv · 2024-03-24

**Q2-1 Originality-Novelty:** 3
**Q2-2 Correctness-Technical Quality:** 3
**Q2-5 Clarity Of Writing:** 3

**Q1 Summary And Contributions:**

The paper proposes an approach to optimizing discounted reward values in finite-state MDPs in the presence of epistemic uncertainty modelled in a Bayesian fashion. This is done in an offline setting given a set of MDP trajectories. There are two steps: the first is to evaluate, separately, the variance due to aleatoric/epistmic uncertainty by analytically solving a fixed, learnt MDP and policy; the second is a stochastic gradient-based method for learning an optimal policy. Experiments shows the benefits on both synthetic benchmarks and a more realistic example from the medical domain.

**Q2-3 Extent To Which Claims Are Supported By Evidence:**

3: Good: the main claims are supported by convincing evidence (in the form of adequate experimental evaluation, proofs, (pseudo-)code, references, assumptions).

**Q2-4 Reproducibility:**

3: Good: key resources (e.g. proofs, code, data) are available and key details (e.g. proofs, experimental setup) are sufficiently well-described for competent researchers to confidently reproduce the main results.

**Q3 Main Strengths:**

+ The problem solved is well motivated and the comparison to related techniques is clearly explained

+ The experimental results are convincing, evaluating both parts of the paper, on a good selection of benchmarks and with a comparison to appropriate baselines

**Q4 Main Weakness:**

- Some aspects of the methods are could be explained more clearly in Section 4 (for example the notion of policy softness is included in Alg 2 but does not become clear until the experimental setup is described; and the usage of Alg 1 within Alg 2 could be presented a bit more clearly)

- The approach is limited to relatively small MDPs (<1000 states) due to the use of matrix inversion for policy evaluation. It would have been good to see some experimental results on exactly how quickly this limit is reached, rather than just a statement of the (cubic) asymptotic complexity.

**Q5 Detailed Comments To The Authors:**

p.3: "can be obtained analytically for a given finite-state MDP" - just for policy evaluation, presumably, not optimisation?

**Q9 Complying With Reviewing Instructions:**

Yes

---

> ### Author Rebuttal · Authors · 2024-04-09
>
> We thank the reviewer for the thorough reading of our manuscript and would like to address here some of the points and questions raised.
>
> *Main weaknesses*
>
> 1) We are thankful to the reviewer for pointing out this possible improvement to the presentation of our method and have edited the manuscript to clarify the usage of policy softness as it is employed in Alg 2, by moving the paragraph describing the initial policies (“in Algorithm 2 and the second order policy, we choose the initial policies to be a softened version of the Nominal policy (η = 0.1). Just like the MLE-optimal policy, the required amount of computation to find these initial policies is one round of value iteration [Sutton and Barto, 2018] and is therefore a computationally negligible addition to the algorithm”) to be immediately after the first mention of Algorithm 2 in the text.
>
> 2) We ran additional computational benchmarks to better quantify the performance of our algorithm with increasing MDP size and report the results in a new figure (available at the link https://figshare.com/s/2c8d101b486a2c2f6e26). The results suggest that our approach may be computationally viable even for MDPs larger than 1000 states with some adjustments.
>
> We plan to include this figure in the Appendix with the following accompanying text.
>
> "Here, we show the increase in compute time for the experiments on synthetic MDPs (results reported in Figures 3 and 4) to compute 1000 gradient steps for different MDP state-space sizes. The red line (batch 256, resampling) corresponds to the same experimental setup, where a new batch of size 256 is sampled at every gradient step. We also present results for a smaller batch size (8) that requires the computation of fewer matrix inverses at each gradient step. Since the sampling itself also presents a significant portion of the computation, we present results where the sampling is only done once at the start of training (no resampling). While sampling a single time is not recommended, the results presented here show the potential computation time that can be saved by resampling from the MDP posterior every n steps rather than at every step. The results suggest that the approach may still be computationally viable for MDPs larger than 1000 states with some adjustments, but further work is necessary to assess the effect these would have on convergence and performance."
>
> *Detailed comments*
>
> Indeed, we mean the value for any given policy can be obtained analytically for policy evaluation.

---

### Meta-Review · Area_Chair_LqzF · 2024-04-15

**Summary:** The paper studies decision-making in a finite-state MDP with unknown dynamics but a number of trajectrories given in a dataset. This is a fairly common setting, and the paper uses Bayesian RL as a method to deal with and capture uncertainty. The main innovation is to follow previous work (in other domains such as Computer Vision and Bayesian Deep Learning) and differentiate between epistemic and aleatoric uncertainty---which in this case can be reduced to estimating the first two moments of the return distribution (using samples from the posterior). Given the two moments, the value of the posterior expected value can be computed analytically, which, in turn, can be used for gradient updates of the policy. The approach is demonstrated on illustrative toy domains, and a slightly larger-scale medical application.

**Recommendation:**  After the rebuttal all reviewers are in favor of accepting the paper (7, 7, 6, 6, 5). Some criticism was raised during the initial reviews and addressed by the authors. Reviewers agree that the problem is timely and interesting and that the approach is sound. The main criticism, which I personally share, is centered around the scalability of the approach, the fairly limited novelty of the method, and all standard issues with policy optimization in the offline setting. The authors have addressed this criticism to a sufficient degree, though it partly remains (at a high level). Overall I think the paper is ready for publication at UAI - I personally estimate it to be in the lower 50% of **accepted** papers (which is a big guess since I only have visibillity over a small batch of submissions) and therefore suggest a poster. I think the main merit of the paper is the discussion of the medical example, including a discussion of the challenges in medical practice, and it might contribute to highlighting the importance of distinguishing between aleatoric and epistemic uncerainty (and how to deal with it in a Bayesian fashion) to the corresponding community.